# Erosive Potential of Sports, Energy Drinks, and Isotonic Solutions on Athletes’ Teeth: A Systematic Review

**DOI:** 10.3390/nu17030403

**Published:** 2025-01-23

**Authors:** Francisca Gálvez-Bravo, Francisca Edwards-Toro, Rafael Contador-Cotroneo, Catalina Opazo-García, Hans Contreras-Pulache, Eliane A. Goicochea-Palomino, Gloria Cruz-Gonzales, Jeel Moya-Salazar

**Affiliations:** 1Asociación de Odontología Deportiva de Chile, AODCH, Santiago 56001, Chile; galvez.fran@gmail.com (F.G.-B.); edwards.toro@gmail.com (F.E.-T.); copazog@uft.edu (C.O.-G.); 2Facultad de Odontología, Universidad Finis Terrae, Santiago 56001, Chile; 3Facultad de Odontología, Universidad de Chile, Santiago 56001, Chile; rcontador@odontologia.uchile.cl; 4Vicerrectorado de investigación, Universidad Norbert Wiener, Lima 51001, Peru; hans.contreras@uwiener.edu.pe; 5Faculties of Health Science, Universidad Tecnológica del Perú, Lima 51001, Peru; elagoichi@gmail.com; 6Faculties of Medical Technology, Universidad Nacional Federico Villareal, Lima 15001, Peru; 7Faculties of Health Science, Universidad Privada del Norte, Lima 51001, Peru

**Keywords:** energy drinks, isotonic solutions, oral health

## Abstract

Background/Objectives: Dental erosion occurs due to repeated contact between the teeth and acidic substances along with mechanical stress. Athletes are exposed to acids through the consumption of sports drinks, energy drinks, and isotonic solutions; they also undergo mechanical stress during training and competition, making them prone to a higher prevalence of dental erosion. Therefore, our aim was to determine the erosive potential of beverages consumed by athletes. Methods: We conducted a systematic review of 1466 articles found on nine search engines between 1997 and 2021. We included observational studies and clinical trials in English, Portuguese, and Spanish on beverage consumption in athletes of both genders. Results: A total of four studies involving 567 athletes from four countries were identified. The prevalence of dental erosion ranged from 19.4% to 100%, and the severity assessments showed that between 52.4% and 75.2% of athletes had enamel affected, and 24% to 57.1% had both enamel and dentin affected. Only one study found that the consumption of sports drinks by swimmers practicing in chlorinated pools doubles the risk of developing dental erosion. Bias was low in half of the studies. Conclusions: The available evidence suggests that the consumption of sports drinks alone is not associated with dental erosion. However, to establish more conclusive evidence on the erosive potential of sports drinks, energy drinks, and isotonic solutions on the oral health of athletes, more prospective cohort studies are needed. These studies should include a standardization of indices and variables to which athletes are subjected, including dietary and healthcare habits, oral conditions, and protective factors. Furthermore, a larger number of athletes must be included to establish more conclusive evidence on the erosive potential of sports drinks, energy drinks, and isotonic solutions on athletes’ oral health.

## 1. Introduction

Athletes are at risk of poor oral health due to some disorders in the oral cavity being related to engaging in a set of sports activities. To achieve optimal performance, athletes involved in competitive sports often consume foods, beverages, sports drinks, energy bars, and gels rich in carbohydrates and energy [1,2]. Although the low pH and acidic nature of energy and sports drinks contribute to tooth erosion, these products are generally marketed without expert guidelines regarding oral health. Frequent carbohydrate intake in the form of sports drinks and beverages may facilitate the growth of salivary cariogenic bacteria (such as *S. mutans*, *Lactobacillus* spp.) and reduce salivary Ig-A levels, thereby rendering athletes susceptible to dental caries. Alterations in saliva composition, therefore, further elevate the risk of poor oral health [2]. In particular, intense exercise can lead to immunodepression, which makes athletes more susceptible to widespread diseases. It has been demonstrated that athletes who engage in rigorous physical training exhibit a diminished immune response and an elevated susceptibility to a range of infections, including periodontitis [1,2]. This has a significant impact on salivary flow, which plays a crucial role in protecting the body from various intraoral pathogens. The reduction in S-IgA production and its correlation with an elevated intraoral growth of pathogenic bacteria highlight the potential for training to act as an “open window” for oral cavity diseases [1]. Evidence suggests a high prevalence of problems in athletes’ oral health, indicating a significant need for treatment as they can lead to negative effects on their training and professional athletic performance [3,4,5].

One study showed that 32.0% of athletes reported an oral health-related impact on sports performance, either due to oral pain, difficulty participating in their usual training and competitions, reduced training volume, or decreased performance [6]. A recent systematic review found that poor oral health is also related to body balance, and cardiorespiratory and cognitive function [5]. Among the most common oral health problems in athletes, the prevalence of caries varies from 15 to 70%, dental trauma from 14 to 70%, and dental erosion from 36% [1] to 47.07% [7]. Therefore, regular inspection and the use of effective oral health promotion strategies can minimize performance impacts caused by poor oral health [6]. 

On the other hand, it is quite clear that consuming fluids before, during, and after exercise minimizes the harmful effects of dehydration on cardiovascular dynamics, body temperature regulation, and athletic performance. Additionally, in certain situations, a sports drink enhances athlete performance, for example, when the athlete has low initial glycogen reserves or the exercise is long (>1 h) at high intensity [8]. Among the beverages consumed by athletes are isotonic and energy drinks, which contain large amounts of sugar in their composition and have an acidic pH lower than 5.5 with the potential to contribute to the development of erosions and dental caries, among others [6,7,8,9]. 

The tooth surfaces that undergo erosion are the enamel (the outermost surface of the tooth) and the dentin (the tissue located beneath the enamel). When a tooth is eroded, the enamel is the initial tissue to be affected. Once all this tissue is worn away, the dentin will then be exposed [10,11,12]. The structural integrity and physical properties of these dental tissues are compromised when they come into contact with acidic substances [10]. The initial stage of this process involves the diffusion of these substances through the acquired film on the tooth surface, a thin acellular biofilm composed of proteins, enzymes, glycoproteins, carbohydrates, and lipids in intimate contact with the dental surface. Once there, hydrogen ions cause the dissolution of the enamel crystals. If this action is repeated over time, coupled with mechanical stress [10,11,12], the tooth surface will soften and subsequently undergo localized, generalized, or asymmetric loss of dental structure [10].

Although erosive wear is partly a physiological process throughout life, it is necessary to distinguish the pathological loss of dental tissues [13] from the components of foods and beverages that may exacerbate the situation. Previous studies have highlighted that pathogenesis can be caused by the use/abuse of different sports and sugary or energy drinks [1,7]. The oral health of athletes is often compromised, with a notably higher risk of dental erosion. Studies have shown that athletes tend to have poor oral health, with a high prevalence of dental erosion ranging from 36% to 85% and dental caries from 15% to 75% [14]. Additionally, a significantly higher prevalence of dental erosion has been reported among athletes (53%) compared to non-athletes (7%) [15]. However, no significant differences were found in the prevalence of dental caries or salivary parameters between these groups [16]. 

Therefore, we aimed to determine the erosive potential of these beverages on athletes’ teeth. This is relevant because oral health problems in athletes have the potential to impact not only their athletic careers but also their overall quality of life. 

## 2. Materials and Methods

### 2.1. Study Design, Data Sources, and Search Strategy

This review follows the guidelines specified in the Preferred Reporting Items for Systematic Reviews and Meta-Analysis (PRISMA) 2020 [17,18] and was registered in PROSPERO (CRD000001). A manual literature search was conducted in September 2022, across six databases (PubMed, Scopus, Web of Science, Scielo, LILACS, and Science Direct), two preprint servers (SportRxiv and Preprints), and Google Scholar (Figure 1). English keywords and their corresponding translations in Spanish and Portuguese were used in the search equation “((erosive potential OR erosive capacity) AND (sports drinks OR energy drinks OR isotonic solutions) AND (dental))” (Table A1).

### 2.2. Inclusion and Exclusion Criteria

The included studies met the following criteria: research involving athletes of both genders between 1997 and 2021, original research articles, brief reports, commentaries, and case series published in English, Portuguese, or Spanish. Excluded studies comprised systematic reviews, meta-analyses, narrative reviews, reflective and historical articles, position papers, books and book chapters, conference proceedings, editorials, methodological papers, letters to the editor, and studies based on secondary data (Table A2).

### 2.3. Screening Study, Data Extraction, and Analysis

Three researchers (F.G., F.E., and R.C.) collected data from scientific databases, applying the inclusion and exclusion criteria. Inter-observer agreements were reviewed in meetings with the research team, and the Kappa concordance test was used. Data extraction was performed using a data matrix in MS Excel 2013 (Microsoft Corp., Redmond, WA, USA), where data from selected studies were tabulated by author, year of publication, study type, type of sport, sample size, sample characterization, type of beverage, examiner characteristics, erosion index, and severity of dental erosion. 

The methodological quality of the manuscripts was evaluated using the Robvis v2.0 tool from Cochrane [19], determining five parameters for each study and an overall quality assessment. Data analysis was conducted in SPSS v24.0 (IBM, Armonk, NY, USA), applying descriptive statistics to estimate frequencies and measures of central tendency (Figure A1).

## 3. Results

The initial search identified 1466 articles. Of these, 800 were from Google Scholar, 559 from Science Direct, 32 from Web of Science, 30 from Scopus, 30 from PubMed, seven from LILACS, and one from Scielo. No reports were found in SportRxiv and Preprints. Duplicated research was discarded, resulting in 1360 articles, and after applying selection criteria, only five studies were included for final reading. One was removed for not meeting post-reading association analysis, finally including four documents [20,21,22,23].

### 3.1. Characteristics of the Studies

The studies were published between 1997 and December 2021, including a total population of 567 athletes (swimmers, cyclists, runners). Each descriptive study was conducted in a different country, specifically in England, the United States, Brazil, and Portugal (Table 1).

The study by Milosevic et al. [20] was the oldest, conducted in 1997 in England, and evaluated 45 amateur athletes, including 25 swimmers and 20 cyclists. They aimed to assess dental status using the Smith and Knight erosion index and sports supplement intake using a beverage exposure questionnaire. Three types of beverages were examined: sports drinks (Isostar, Dextrain), energy drinks (Lucozade, High 5, PSP22), and isotonic solutions (unspecified brand). They found that all participants had some degree of erosion, and although the severity differed, dentin erosion was significantly higher in cyclists compared to swimmers.

Mathew et al. [21] conducted a study in 2002 involving a total of 304 university athletes from different disciplines, excluding swimmers to avoid confounding exposure to chlorinated water. They evaluated whether regular consumption of sports beverages was associated with dental erosion using the Lussi index and a custom questionnaire. Sports beverages (Carbofuel) and isotonic solutions (Gatorade, Powerade, Allsport) resulted in three out of ten participants experiencing some degree of erosion. Moreover, the majority showed erosion in the enamel, with some cases extending to both enamel and dentin.

Antunes et al. [22] conducted their research in 2017 on 108 Brazilian runners, assessing the presence or absence of erosion with the Eccles index and potential risk factors for dental erosion with a questionnaire analyzing sports beverages and isotonic solutions. They found that nearly two out of every ten participants experienced some degree of erosion, predominantly affecting enamel and dentin, followed by erosion limited to the enamel.

Lastly, Silva et al. [23] conducted their study in 2021 on 110 Portuguese athletes, including 55 swimmers and 55 non-swimmers (bodybuilders, football players, boxers, volleyball players, runners). Using the basic erosive wear examination index (BEWE), they evaluated the presence of dental erosion and its relationship with the consumption of sports beverages (Isostar, Gold Nutrition), energy drinks (Monster, Redbull), and isotonic solutions (Gatorade, Powerade). Their results showed that eight out of ten participants experienced some degree of dental erosion, although they did not specify the severity level.

The four studies were also heterogenous with regards to the beverages reported and the indices used to measure or evaluate dental erosion (Table 2). Each study used different indices to measure or evaluate dental erosion. Heterogeneity was found in the data obtained regarding beverage exposure registration, where only one study did not report it [22]. 

The prevalence of dental erosion varied from 19.4% to 100%. Similarly, the severity assessments showed that 36% had erosion on tooth surfaces, 85% on posterior teeth, and between 52.4% and 75.2% on enamel and 24% and 57.1% on enamel and dentin. Regarding the clinical site of the findings, two studies showed a significant degree of erosion by dividing teeth into sextants. Milosevic et al. [20] presented results in the anterior sextant, and Silva et al. [23] indicated that, excluding third molars, each sextant had the greatest loss of hard tissue. 

The other two studies showed that mandibular molars were the most affected. In the case of Mathew et al. [21], these presented the most severe and extensive erosion, with the first mandibular molar being the most affected and its occlusal surface being the most affected dental surface. Likewise, Antunes et al. [22] found that the erosion rate in the mandible was 23.8% in anterior teeth and 61.9% in posterior teeth. This was followed by maxillary teeth with a rate of 4.8% in anterior teeth and 28.6% for posterior teeth (Table 3).

### 3.2. Dental Erosion and Sports Beverages

Regarding the relationship between dental erosion and beverage consumption, the studies conducted by Milosevic et al., Mathew et al., and Antunes et al. concluded that one-third of the participants had dental erosion. However, the consumption of sports beverages alone was not associated with the presence or development of dental erosion. On the other hand, Silva et al. [23] reported that the consumption of energy drinks by swimmers in chlorinated pools doubled the risk of developing dental erosion compared to athletes who were not swimmers but consumed the same beverages. Additionally, as a sub-analysis, Antunes et al. [22] found a significant association between dental erosion and gastroesophageal reflux. Moreover, in the case of runners, erosion was significantly associated with the frequency of races per week (*p* = 0.04) and the time spent on competition (*p* = 0.01) (Table 4).

### 3.3. Bibliometric and Methodological Quality Analysis

Regarding affiliated institutions in the four studies, we identified that Mathew et al. [21] conducted their study at The Ohio State University, and Silva et al. [23] at the Faculty of Health Sciences, University Fernando Pessoa, Portugal. Meanwhile, Milosevic et al. [20] had the support of the University of Liverpool to evaluate athletes training in two public pools in Liverpool and three cycling clubs in northwest England. Antunes et al. [22] obtained ethical approval from Fluminense Federal University for the participation of athletes attending events in the city of Nova Friburgo, Rio de Janeiro (Brazil).

As for funders, only Mathew et al. [21] received research grant support from Quaker Oats Company, Chicago, IL, USA. Additionally, none of the included studies provided openly accessible data in any public repository or Appendix A. Bias analysis identified two studies with low risk of bias and two studies where bias was indeterminate (Figure 2).

## 4. Discussion

This systematic review analyzed four studies from four countries and 567 athletes. We found that the development of dental erosion is not related to the consumption of sports beverages, energy drinks, and isotonic solutions. Although the methodological quality is moderate and the study characteristics did not allow an in-depth analysis, this research explores this relationship with the available evidence.

### 4.1. Strengths

The strength of this study is the consolidation and review of the available data from clinical trials on the erosive potential of sports beverages, energy drinks, and isotonic solutions on athletes’ teeth. Often, most studies tend to focus on studying their overall dental health and the impact it may have on their training and athletic performance [2,3,5]. Moreover, certain in vitro studies use animal teeth such as bovines [21] or non-erupted human teeth [22] instead of studying them directly in the human population.

### 4.2. Main Findings

In the human population, research on the impact of consuming sports beverages, energy drinks, and isotonic solutions on athletes’ dental health is still limited [20,21,22,23]. Previous studies have suggested that athletes, particularly those who consume large quantities of sports drinks, may be at increased risk for dental erosion due to multiple factors [1]. However, because this was a narrative review without clearly defined inclusion criteria or a systematic bibliographic search, the association between sports drink consumption and the incidence and severity of dental erosion in athletes remains unclear. To address this issue, our systematic review, which analyzed 1360 documents, found no significant relationship between the consumption of these drinks and dental erosion. These findings may be explained by variations in study design, sample sizes, and differences in athlete characteristics (i.e., type of sport, training duration, and frequency of dental monitoring) [24,25]. Future research should account for these variables to provide a more accurate assessment of the relationship between sports drinks and oral health.

Several in vitro studies have shown that the acidic solutions present in these beverages can cause erosion of dental enamel. For example, using samples of bovine teeth, it has been found that citric acid, the most common acid used in sports beverages, has a greater erosive effect compared to other acids such as malic acid [26]. Additionally, in non-erupted human molars studied under controlled conditions, it was observed that enamel erosion by dietary acid solutions is influenced by the interaction of pH, acid concentration, and the presence of calcium [26]. It is well known that energy drinks are potentially erosive to teeth because they contain large amounts of sugar and have an acidic pH lower than 5.5 [9]. Although the prevalence of dental erosion found in the studies included in our review varied from 19.4% to 100%, they did not report a relationship between erosion and the consumption of sports beverages, energy drinks, and isotonic solutions.

Dental erosion can be etiologically distinguished from mechanically induced defects such as abrasion and attrition. However, these phenomena often overlap clinically and make precise distinctions challenging. Based on the origin of the acids causing erosion, a distinction is made between intrinsic and extrinsic erosions. The former typically affects mainly palatal and occlusal dental surfaces, while extrinsically triggered erosion begins on the vestibular surfaces of anterior teeth [27]. In the studies by Milosevic et al. [20], Mathew et al. [21], and Antunes et al. [22], the incisal edges of anterior teeth (a subset of the vestibular surface of anterior teeth) are excluded due to the difficulty in diagnosing erosion exclusively from abrasion and attrition, which could lead to the underdiagnosis of dental erosion in the studies.

On the other hand, an aspect not considered in the four studies is the exposure to protective factors that could alter the obtained results, such as fluoride, as its buffering capacity and concentration are inversely correlated with erosive potential, potentially having significant effects against enamel erosion [28]

It is important to highlight that saliva is the main biological factor influencing the progression of dental erosion [29]. Among the properties of saliva to consider are its flow, buffering capacity, pH, and composition [30]. Therefore, understanding its components and properties related to its protective function can drive the development of preventive measures to enhance its positive effects [29]. Only in the study by Milosevic et al. [20] was salivary flow evaluated, concluding that exposure to a sports beverage (Maxim^®^) for 1 min followed by rinsing with water may decrease salivary flow, which is counterproductive and inconclusive.

Previously, Coombes in 2005 found no clear association between dental wear and sports beverage consumption [8]. This suggests that isolating a particular dietary component may be insufficient and that factors such as consumption habits and saliva production may play a more significant role in this issue. Sports beverages do not contain more acids than a wide variety of beverages such as soft drinks, fruit juices, beers, and wine. Further research is needed to clearly understand the role of saliva in high-performance athletes and to associate its effects with dental problems.

One study found that over 26% of competitive swimmers and 10% of recreational swimmers had dental erosion due to exposure to chlorinated pools. Erosions in competitive swimmers were on the vestibular and palatal surfaces of anterior teeth, while recreational swimmers developed them exclusively on the palatal surfaces. Although the pool water pH was neutral, it was subsaturated with respect to hydroxyapatite, leading to demineralization and a decrease in local pH and other components on the tooth surface, causing enamel dissolution [31]. Another study in India [32] found that 48.2% of professional swimmers suffered dental erosion, mainly affecting the palatal surfaces of upper anterior teeth followed by the lingual surfaces of lower anterior teeth. This contradicts the etiology of erosion, where external factors mainly damage the vestibular surfaces of anterior teeth and intrinsic factors such as gastroesophageal reflux affect palatal and occlusal surfaces. Although the studies by Milosevic et al. [20] and Silva et al. [23] also included swimmers, they did not declare how exposure to chlorinated pools affects them, so it could act as a confounding factor. This is confirmed by the study by Mathew et al. [21], who decided to exclude them to avoid confusing exposure to chlorinated water.

There are underlying or concomitant oral conditions that predispose the patient to have hard tissue defects, such as amelogenesis and dentinogenesis imperfecta. It is imperative to consider the patients’ significant expression levels of genotype and phenotype. To identify a possible cofactor in the development of dental erosion, further investigation of gene-specific relationships may provide valuable clarification in future studies [33]. 

### 4.3. Limitations

The most significant limitation of this review is the lack of detailed examination of confounding variables such as personal oral hygiene practices, dietary habits, and dental check-ups, all of which could mitigate the erosive potential of acidic beverages. Future research should incorporate these variables to develop and promote preventive oral health programs for athletes [34]. Additionally, the review did not consider exposure to fluoride, which has a buffering capacity inversely related to erosive potential [26], nor did it account for exposure to chlorinated pools, which can cause enamel dissolution in swimmers [28]. Finally, the limited number of studies on this topic prevented a meta-analysis, thus limiting the power and precision of the findings [14].

## 5. Conclusions and Future Directions

Current evidence suggests that there is no clear association between sports beverage consumption alone and dental erosion. However, to draw more definitive conclusions about the erosive impact of sports beverages, energy drinks, and isotonic solutions on athletes’ oral health, larger controlled prospective studies are required. These studies should also consider factors such as the frequency and duration of beverage consumption, the interaction with oral hygiene habits, and the type of sports activity performed. Additionally, exploring the effectiveness of preventive measures, such as modifying the composition of beverages or using specific mouth rinses, would be beneficial in reducing the risk of dental erosion in athletes.

We suggest using in situ study models, where participants wear a removable device with small portions of extracted teeth that they would wear for a given period while consuming sports beverages, energy drinks, and isotonic solutions, so we can replicate and evaluate the effects on teeth without harming the participants’ teeth. It is essential to standardize the indices and variables to which athletes are subjected, including dietary and healthcare habits, oral conditions, and protective factors. Subsequently, these should be subjected to micro-CT and microhardness studies on the tooth portions. 

## Figures and Tables

**Figure 1 nutrients-17-00403-f001:**
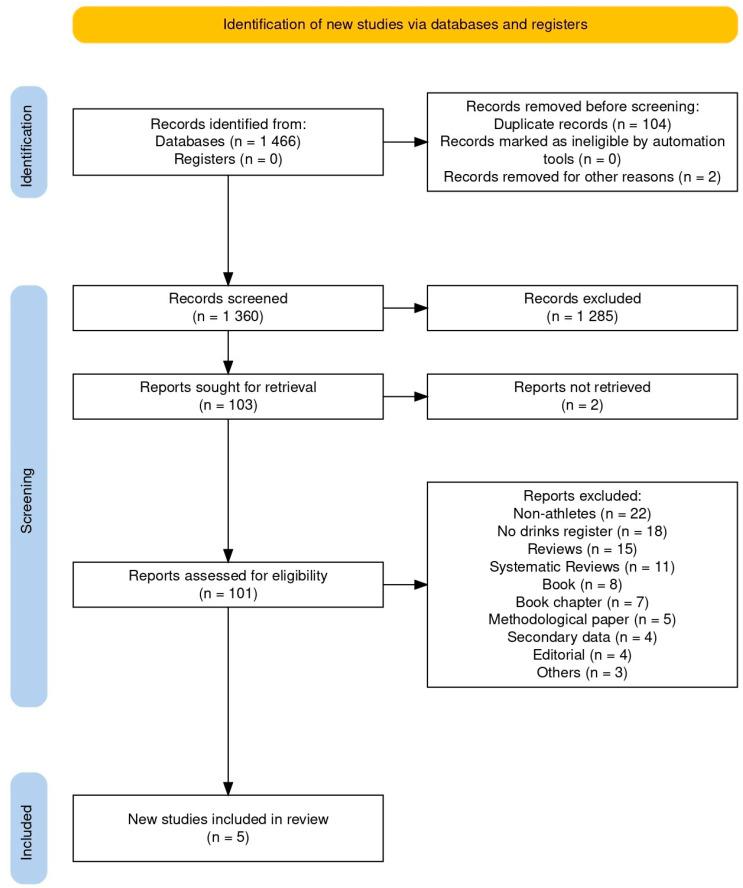
Flowchart of the article selection process for the review.

**Figure 2 nutrients-17-00403-f002:**
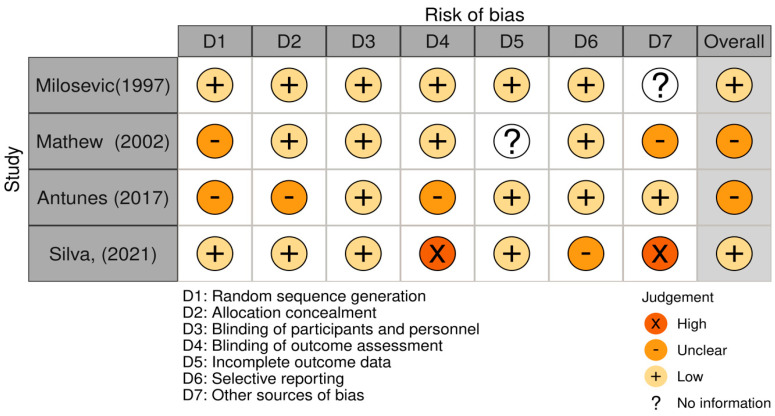
Analysis of bias in the included studies [20,21,22,23].

**Table 1 nutrients-17-00403-t001:** Selected studies.

Author (Year)	Country	Type of Study	Type of Sport	Sample Size	Journal	Publisher	Impact Factor (SJR)	H-Index	Quartile
Milosevic (1997) [20]	England	Descriptive with analytical components	Swimmers and cyclists	45 amateur athletes	British Dental Journal	Nature Publishing Group	Not available, of the year 1997	91	Not available, of the year 1997
Mathew (2002) [21]	United States	Descriptive with analytical components	University athletes	304 amateur athletes	Caries Research	S. Karger AG	0.784	107	Q2
Antunes (2017) [22]	Brazil	Descriptive with analytical components	Runners	108 amateur athletes	Journal of Oral Science	Nihon University, School of Dentistry	0.397	56	Q3
Silva (2021) [23]	Portugal	Descriptive with analytical components	Swimmers and non-swimmers (bodybuilders, soccer players, boxers, volleyball players, runners)	110 amateur athletes	Science and Sports	Elsevier Masson s.r.l.	0.259	27	Q4

**Table 2 nutrients-17-00403-t002:** Characteristics of beverages and consumption pattern.

Author, Year	Sample Size	Studied Beverages		
		Sport Drink	Energetic Drink	Isotonic Solution
Milosevic, 1997 [20]	45 amateur athletes	Isostar, Dextrain	Lucozade, High 5, PSP22	-
Mathew, 2002 [21]	304 amateur athletes	Carbofuel	-	Gatorade, Powerade, Allsport
Antunes, 2017 [22]	108 amateur athletes	No brand	-	Uses sports drinks and isotonic drinks as synonymous
Silva, 2021 [23]	110 amateur athletes	Isostar, Gold Nutrition	Monster, Redbull	Gatorade, Powerade

**Table 3 nutrients-17-00403-t003:** Characteristics related to dental erosion.

Author (Year)	Type of Dental Erosion Examiner	Erosion Index	Exposure to the Beverage	Erosion Prevalence	Severity	Clinical Finding Site
Milosevic, 1997 [20]	Two calibrated examiners	Smith and Knight Erosion Index	1. How long have you been consuming sports drinks? - 70% of cyclists >1 year - 32% of swimmers >1 year 2. How much do you consume per week? - 30% of cyclists > 9 litres - 52% of swimmers > 1 up to 3 litres 3. What concentration do you use when training? - 65% of cyclists 10% concentration - 56% of swimmers 10% concentration 4. What concentration do you use when competing? - 45% of cyclists 10% concentration - 28% of swimmers 10% concentration	100%	36% of swimmers had erosion on dentin surfaces. 85% of cyclists had dentinal erosion on posterior teeth. Dentin erosion was significantly higher in cyclists compared to swimmers (*p* < 0.01).	Cyclists had more palatal erosion than swimmers (*p* < 0.001). Upper posterior and lower vestibular erosion was high in both groups but was present in the anterior sextant too.
Mathew, 2002 [21] *	Two calibrated examiners	Lussi index	1. At least once a week 2. During each training session - 63% reported that they drank at least 1 L of sports drink daily 3. No consumption: 12.3%	36.5% have some degree of erosion	Of the 36.5%, 75.2% had enamel erosion, and 24.8% had erosion in both enamel and dentin.	The tooth most frequently affected by dental erosion was the permanent mandibular first molar, and the most affected dental surface was the occlusal surface of this tooth. The most severe and extensive erosion was also found in the mandibular teeth. 2.3% presented vestibular erosion, 35.5% occlusal erosion, and 0.7% lingual erosion.
Antunes, 2017 [22] *	Trained and calibrated evaluator.	Presence/absence of erosion, Eccles index	Does not say	19.4%	Of the 19.4%, 52.4% had enamel erosion only, and 57.1% had erosion in enamel and dentin.	In the maxilla, the rate was 4.8% for anterior teeth and 28.6% for posterior teeth. In the mandible, the rates were 23.8% and 61.9%, respectively.
Silva, 2021 [23]	Trained and calibrated evaluator.	Basic erosive wear examination index (BEWE)	1. Domestic consumption: 16.4% 2. At least once a day: 3.6% 3. At least once a week: 20% 4. During each training session: 11.8% 5. No consumption: 64.5%	83.6% have some degree of erosion	Does not say	The prevalence of erosive tooth wear was found to be significantly higher in sextants 2 and 5, at 69.1% and 59.1%, respectively.

* Some degree of erosion included some enamel erosion.

**Table 4 nutrients-17-00403-t004:** Results and sub-analysis of the selected studies.

Author	Results	Subanalysis
Milosevic, 1997 [20]	The Spearman’s correlation coefficient was applied to find the relationship between the use of sports drinks and COPD values and dental erosion. No association was found between erosive wear and sports drink consumption. (No further detailed information available).	
Mathew, 2002 [21]	The multiple regression analysis and Spearman’s correlation analysis did not reveal any relationship between any of the factors related to sports drinks and dental erosion.	
Antunes, 2017 [22]	The chi-square test or Fisher’s exact test was used, revealing that sports drink consumption was not associated with dental erosion.	Dental erosion in amateur runners was significantly associated with the frequency of running per week (*p* = 0.04) and the time spent competing (*p* = 0.01). Analysis of potential risk factors related to runners’ daily routine showed no statistically significant difference in the relationship with the prevalence of dental erosion. Analysis of possible systemic risk factors revealed that gastroesophageal reflux was significantly associated with dental erosion (*p* = 0.05).
Silva, 2021 [23]	The consumption of energy drinks by swimmers doubles the likelihood of dental erosion compared to non-swimmer athletes who consume the same energy drinks.	In terms of the risk of dental erosion, swimmers who consume energy drinks represent the highest risk group, followed by athletes (non-swimmers) who consume energy drinks.

## Data Availability

The data in this study are secondary and are available in the repositories of each journal.

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
