# Peer review of "Erosive Potential of Sports, Energy Drinks, and Isotonic Solutions on Athletes’ Teeth: A Systematic Review"

_nutrients, 2025, doi:10.3390/nu17030403_

Round 1

Reviewer 1 Report

Comments and Suggestions for Authors

1. Primary research questionThe main focus of the study is the dental assessment of the erosive potential of beverages frequently consumed by athletes.

2. New or relevant aspects for the field

The study is characterised by a systematic review of the existing literature. The inclusion of observational and clinical studies from different languages expands the scope and depth of the study.

3. Contribution to the field

The need for more rigorous prospective cohort studies to further investigate erosive effects is highlighted. Attention is also drawn to preventive measures to reduce the risk of dental erosion in athletes.

4. Improvements to the methodological approach

A possible improvement to the methodology could be a detailed examination of confounding variables such as personal oral hygiene practices, dietary habits and dental monitoring. However, this work would exceed the scope of a review. However, it would still be an important point of discussion as a limitation of the study.

5. Consistency of the conclusions with the evidence

The study states that although the available data do not establish a clear association between sports drink consumption and dental erosion, further research is needed.

Another important point of discussion that is missing is the concomitant oral diseases that excessive consumption can exacerbate, such as amelogenesis or dentinogenesis imperfecta. I therefore recommend including the following work in the discussion and discussing such concomitants as a cofactor.

Möhlhenrich, S.C., Chhatwani, S., Schmidt, P. et al. Orthodontic findings and treatment need in patients with amelogenesis imperfecta: a descriptive analysis. Head Face Med 20, 36 (2024).

6. Appropriateness of the references

The references appear appropriate, see above.

7. General comments on tables and figures

Appropriate.

Author Response

Comment 1: Primary research question The main focus of the study is the dental assessment of the erosive potential of beverages frequently consumed by athletes.

Response 1: Thank you for pointing this out. We agree with this comment.

Comment 2: New or relevant aspects for the field

The study is characterised by a systematic review of the existing literature. The inclusion of observational and clinical studies from different languages expands the scope and depth of the study.

Response 2: We agree with this comment.

Comment 3: Contribution to the field

The need for more rigorous prospective cohort studies to further investigate erosive effects is highlighted. Attention is also drawn to preventive measures to reduce the risk of dental erosion in athletes.

Response 3: We agree with this comment.

Comment 4: Improvements to the methodological approach

A possible improvement to the methodology could be a detailed examination of confounding variables such as personal oral hygiene practices, dietary habits and dental monitoring. However, this work would exceed the scope of a review. However, it would still be an important point of discussion as a limitation of the study.

Response 4: Thank you for pointing this out. We agree with this comment. Therefore, the suggested change was made based on the reviewers' comments. [page 10, paragraph 5, line 297 - 307]

Comment 5: Consistency of the conclusions with the evidence

The study states that although the available data do not establish a clear association between sports drink consumption and dental erosion, further research is needed.

Another important point of discussion that is missing is the concomitant oral diseases that excessive consumption can exacerbate, such as amelogenesis or dentinogenesis imperfecta. I therefore recommend including the following work in the discussion and discussing such concomitants as a cofactor.

Möhlhenrich, S.C., Chhatwani, S., Schmidt, P. et al. Orthodontic findings and treatment need in patients with amelogenesis imperfecta: a descriptive analysis. Head Face Med 20, 36 (2024).

Answer 5: Thank you for pointing this out. We agree with this comment. Therefore, the suggested change was made based on the reviewers' comments. [page 10, paragraph 4, line 291 - 206]

Comment 6. Appropriateness of the references The references appear appropriate, see above.

Answer 6: Thank you for pointing this out. We agree with this comment.

Comment 7. General comments on tables and figures

Answer 7: Thank you for pointing this out. We agree with this comment.

Reviewer 2 Report

Comments and Suggestions for Authors

Dear authors, thank you for submitting the manuscript "Erosive potential of sports, energy drinks, and isotonic solutions on teeth: a systematic review". I enjoyed reading your paper and here is my feedback:

-Create your hypotheses and they are usually created by information not used in the meta-analysis.

-Combine some paragraphs so they look bigger. Right now you have some paragraphs with two sentences such as 2 and 3.

-In the third paragraph for treating dental trauma while performing sports, I recommend you this reference PMID: 33363807.

-For the erosive wear being part of the physiological process, please be more specific of the regular amount in mm or % for this process.

-Revise the grammar for the 2.2 section, specifically which were included or excluded.

-Create a small table describing the exclusion and inclusion criteria.

-You need to expand the limitations of the study.

-Mention what future studies would you like to create based on the results of this review.

-Make conclusion shorter and more general.

Author Response

Comment 1: Create your hypotheses and they are usually created by information not used in the meta-analysis.

Response 1: Thank you for pointing this out. We agree with this comment.

Comment 2: Combine some paragraphs so they look bigger. Right now you have some paragraphs with two sentences such as 2 and 3.

Response 2: Thank you for pointing this out. We agree with this comment. Therefore, the suggested change was made based on the reviewers' comments. [page 1, paragraph 3, line 45 - 51]

Comment 3: In the third paragraph for treating dental trauma while performing sports, I recommend you this reference PMID: 33363807.

Response 3: Thank you for pointing this out. We agree with this comment. Therefore, the suggested change was made based on the reviewers' comments. [page 2, paragraph 1, line  51 -55]

Comment 4: For the erosive wear being part of the physiological process, please be more specific of the regular amount in mm or % for this process.

Response 4: Thank you for pointing this out. It has been considered to report in percentage since the information we have uses that measure.

Comment 5: Revise the grammar for the 2.2 section, specifically which were included or excluded.

Response 5: Thank you for pointing this out. We agree with this comment. Therefore, the suggested change was made based on the reviewers' comments. [page 2, paragraph 6, line 92 - 99]

Comment 6: Create a small table describing the exclusion and inclusion criteria.

Response 6: Thank you for pointing this out. We agree with this comment. Therefore, the suggested change was made based on the reviewers' comments. [page 3, paragraph 1, line  98-99, Appendix A: 346 - 348]

Comment 7: You need to expand the limitations of the study.

Response 7: Thank you for pointing this out. We agree with this comment. Therefore, the suggested change was made based on the reviewers' comments. [page 10, paragraph 5, line 297 - 306]

Comment 8: Mention what future studies would you like to create based on the results of this review.

Response 8: Thank you for pointing this out. We agree with this comment. Therefore, the suggested change was made based on the reviewers' comments. [page 11, paragraph 2, line 318 - 324]

Comment 9: Make conclusion shorter and more general.

Response 9: Thank you for pointing this out. We agree with this comment. Therefore, the suggested change was made based on the reviewers' comments. [page 10-11, line 309 - 317]

Round 2

Reviewer 1 Report

Comments and Suggestions for Authors

my comments were addressed

Author Response

Thank you for your comments

Reviewer 2 Report

Comments and Suggestions for Authors

Dear authors, thank for the major modifications provided to the manuscript. You have fulfilled all my requests, therefore I recommend its publication.

Author Response

Thank you for your comments